# HQA-VLAttack: Towards High Quality Adversarial Attack on Vision-Language Pre-Trained Models

**Han Liu**
Dalian University of Technology
Dalian, China
liu.han.dut@gmail.com

**Jiaqi Li**
Dalian University of Technology
Dalian, China
li.jiaqi.dut@gmail.com

**Zhi Xu**
Dalian University of Technology
Dalian, China
xu.zhi.dut@gmail.com

**Xiaotong Zhang**[*]
Dalian University of Technology
Dalian, China
zxt.dut@hotmail.com

**Xiaoming Xu**
Dalian University of Technology
Dalian, China
xu.xm.dut@gmail.com

**Fenglong Ma**
The Pennsylvania State University
Pennsylvania, USA
fenglong@psu.edu

**Yuanman Li**
Shenzhen University
Shenzhen, China
yuanmanli@szu.edu.cn

**Hong Yu**
Dalian University of Technology
Dalian, China
hongyu@dlut.edu.cn

## Abstract

Black-box adversarial attack on vision-language pre-trained models is a practical and challenging task, as text and image perturbations need to be considered simultaneously, and only the predicted results are accessible. Research on this problem is in its infancy, and only a handful of methods are available. Nevertheless, existing methods either rely on a complex iterative cross-search strategy, which inevitably consumes numerous queries, or only consider reducing the similarity of positive image-text pairs but ignore that of negative ones, which will also be implicitly diminished, thus inevitably affecting the attack performance. To alleviate the above issues, we propose a simple yet effective framework to generate high-quality adversarial examples on vision-language pre-trained models, named HQA-VLAttack, which consists of text and image attack stages. For text perturbation generation, it leverages the counter-fitting word vector to generate the substitute word set, thus guaranteeing the semantic consistency between the substitute word and the original word. For image perturbation generation, it first initializes the image adversarial example via the layer-importance guided strategy, and then utilizes contrastive learning to optimize the image adversarial perturbation, which ensures that the similarity of positive image-text pairs is decreased while that of negative image-text pairs is increased. In this way, the optimized adversarial images and texts are more likely to retrieve negative examples, thereby enhancing the attack success rate. Experimental results on three benchmark datasets demonstrate that HQA-VLAttack significantly outperforms strong baselines in terms of attack success rate.

---

[*]Corresponding author.

39th Conference on Neural Information Processing Systems (NeurIPS 2025).

# 1  Introduction

Vision-Language Pre-training (VLP) models have become a cornerstone for cross-modal tasks, achieving remarkable success in applications such as image-text retrieval [39, 4, 7], image captioning [27], and visual grounding [20]. However, research has shown that these models are vulnerable to adversarial attacks [19, 10, 37, 6, 11], posing significant societal concerns. Adversarial attacks inject imperceptible perturbations to text and image inputs, aiming to manipulate predictions of victim VLP models maliciously. Specifically, existing attacks can be broadly categorized into white-box attacks [37, 21, 32] and black-box attacks [19, 10, 34, 14, 5]. In white-box attacks, attackers have full access to the victim model, allowing them to exploit gradients for highly effective attacks. However, the white-box setting can be too idealistic in real-world scenarios. In contrast, black-box attacks assume limited access to the victim model, such as confidence scores or prediction labels, making them more practical for real-world applications.

Black-box attacks can be categorized into **query-based attacks** [34, 14, 5, 18, 17] and **transfer-based attacks** [19, 10, 35, 38]. Query-based attacks employ an iterative cross-search strategy that requires repeatedly querying the victim model and utilizing its feedback to refine adversarial perturbations. While effective, these methods incur substantial query costs, limiting their practicality in real-world applications. In contrast, transfer-based attacks generate adversarial examples by optimizing them on a surrogate model, leveraging feature similarity and generalization to maintain their effectiveness against unseen victim models without requiring queries. Due to their independence from direct access to the victim model, transfer-based attacks are particularly well-suited for real-world adversarial scenarios, making their enhancement a critical research focus.

As shown in Figure 1, existing transfer-based adversarial attacks on VLP models primarily aim to decrease the similarity of positive image-text pairs, thus causing the victim model to retrieve more negative examples and improving the attack success rates. Specifically, SGA [19] utilizes BERT-Attack for text perturbation along with set-level guidance to explicitly reduce the similarity between positive image-text pairs. In addition, DRA [10] enhances this approach by incorporating trajectory-aligned diversified sampling and text-guided selection, which maximizes the semantic distance between positive image-text pairs, further increasing the attack success rate. However, both SGA and DRA primarily focus on reducing the similarity of positive image-text pairs, inadvertently reducing the similarity of negative pairs as well. Consequently, VLP models may still be biased towards retrieving positive examples, which can inevitably diminish the overall attack success rates. For further experimental details regarding Figure 1, please refer to Appendix A.

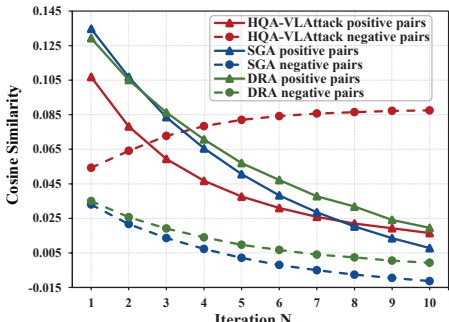

Figure 1: The average cosine similarity of image-text pairs optimized by SGA, DRA, and HQA-VLAttack on the Flickr30K dataset using ALBEF as the surrogate model.

To address the limitation mentioned above, we propose a **H**igh **Q**uality transfer-based **A**dversarial **V**ision-**L**anguage **Attack**, namely **HQA-VLAttack**. The overview of HQA-VLAttack is depicted in Figure 2. By "high quality", we mean that HQA-VLAttack achieves a significantly higher attack success rate compared to existing methods. Specifically, HQA-VLAttack first generates semantically consistent adversarial texts, followed by adversarial images with low similarity to the original images. Finally, contrastive learning is used to increase the distance between matched adversarial image-text pairs while reducing the distance between unmatched pairs. Experimental results on three datasets and three attack tasks demonstrate that HQA-VLAttack outperforms other strong baselines, establishing it as an effective high-quality vision-language adversarial attack method[2].

---

[2]The source code is publicly available at `https://github.com/HQA-VLAttack/HQA-VLAttack`

## 2 Related Work

### 2.1 White-Box Adversarial Attacks on VLP Models

White-box adversarial attacks [37, 32] assume that the attackers have full access to all information about the victim model, including its architecture, training data, and gradients. This enables attackers to directly exploit gradients and generate highly effective adversarial perturbations. Co-Attack [37] extends this paradigm to multimodal settings by simultaneously perturbing both image and text modalities, thereby leveraging intermodal dependencies to generate more effective adversarial examples. Its collaborative framework overcomes the limitations of single-modal attacks and significantly increases the attack success rate in various vision-language tasks. However, the white-box assumption can be too idealistic in real-world scenarios, as most developers will not release model details to the public. This significantly limits the practical deployment of such attack methods.

### 2.2 Black-Box Adversarial Attacks on VLP Models

Black-box adversarial attacks restrict access to limited model output, such as confidence scores or predicted labels, making them more practical for real-world applications. When targeting VLP models, these attacks can be broadly categorized into query-based and transfer-based attacks.

**Query-based attacks** typically employ a complex iterative cross-search strategy on both image and text inputs, leading to high query consumption. VLAttack [34] is one of the most advanced query-based attacks, achieving state-of-the-art performance in attack success rate. It generates adversarial examples by first independently perturbing images and texts, and then refining the adversarial pair through an iterative cross-search strategy that jointly optimizes the multimodal embedding. Although this method achieves effective attack performance, its reliance on a large number of queries significantly limits real-world applicability.

**Transfer-based attacks** generate adversarial examples on a surrogate model and transfer them to deceive the victim model, thereby eliminating the need for queries. This makes transfer-based methods more practical for real-world applications. SGA [19] leverages set-level attacks by generating adversarial examples from multi-scale images and multiple matching captions, thereby enhancing cross-modal interactions and transferability. DRA [10] is a recent transfer-based method, which enhances transferability by diversifying adversarial examples along the intersection region of the adversarial trajectory and incorporates text-guided selection to mitigate overfitting. Most of these methods also use input transformation techniques [31] to further improve transferability. However, current methods lack consideration for negative image-text pairs, causing biased retrieval in VLP models and limiting attack success rates. To address this issue, we propose a high-quality transfer-based attack that reduces the similarity of positive pairs while increasing the similarity of negative pairs. This approach makes VLP models more likely to retrieve negative examples, thereby improving attack performance.

## 3 Problem Formulation

In image-to-text retrieval (TR) task, the retrieval function $F_{TR}$ takes an input image $v_i$. The retrieval model $F_{TR}$ retrieves the top-$k$ candidate texts from a text set $D_t = \{t_1, \ldots, t_n\}$, where $n$ denotes the set size. This retrieval process can be formulated as $F_{TR}(v_i, D_t)_k = \{t^{(1)}, \ldots, t^{(k)}\}$.

In text-to-image retrieval (IR) task, the retrieval model $F_{IR}$ takes an input text $t_i = \{x_1, \ldots, x_L\}$, where $L$ denotes the text length, and retrieves the top-$k$ candidate images from an image set $D_v = \{v_1, \ldots, v_n\}$, where $n$ denotes the set size. Mathematically, this retrieval procedure can be expressed as $F_{IR}(t_i, D_v)_k = \{v_i^{(1)}, \ldots, v_i^{(k)}\}$.

Transfer-based multimodal adversarial attacks aim to generate image and textual adversarial example sets, $D_v' = \{v_1', \ldots, v_n'\}$ and $D_t' = \{t_1', \ldots, t_n'\}$, by applying pixel-level perturbations to images and word-level perturbations to texts using a surrogate model $f_\phi$. Here, we use $(v_i', t_i')$ to represent a positive adversarial image-text pair. Then a successful attack on the TR task can be formulated as:

$$t_i' \notin F_{TR}(v_i', D_t')_k \quad \text{s.t.} \quad \|v_i' - v_i\|_\infty \le \epsilon_v, \, d(t_i', t_i) \le \epsilon_t, \tag{1}$$

where $\|\cdot\|_\infty$ denotes the $L_\infty$ norm. The parameter $\epsilon_v$ represents the maximum allowable perturbation for images, while $d(\cdot, \cdot)$ measures the distance between adversarial and original texts, with $\epsilon_t$ denoting

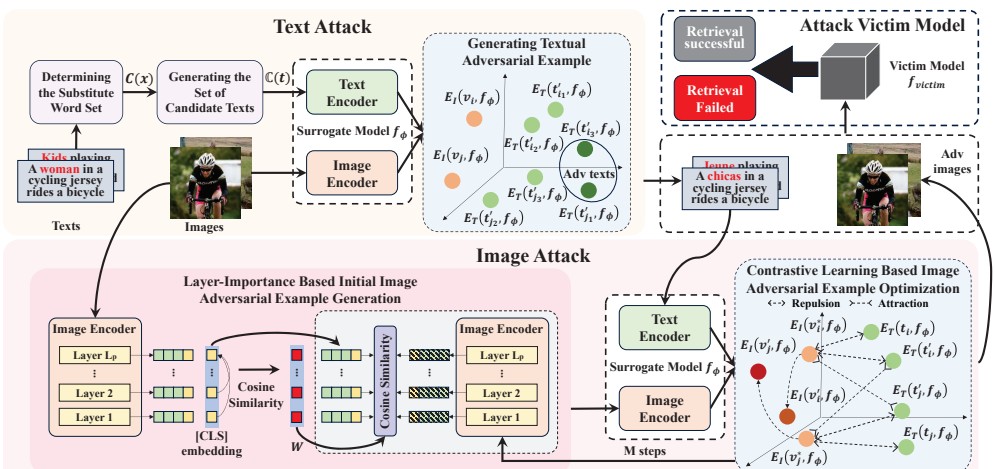

Figure 2: The overall of HQA-VLAttack. First, the Text Attack module determines the substitute word set and then generates the textual adversarial example $t_i'$. Second, the Image Attack module applies layer-importance based initial image adversarial example generation to obtain the initial adversarial image example $v_i'$, followed by contrastive learning-based image adversarial example optimization for further refinement. Finally, the optimized adversarial examples are fed into the victim model.

the maximum allowable perturbation for texts. Similarly, for the IR task:

$$v_i' \notin F_{IR}(t_i', D_v')_k \quad \text{s.t.} \quad \|v_i' - v_i\|_\infty \le \epsilon_v,\, d(t_i', t_i) \le \epsilon_t. \tag{2}$$

These conditions ensure that adversarial examples effectively degrade retrieval performance.

## 4 The Proposed Method

### 4.1 Text Attack

#### 4.1.1 Determining the Substitute Word Set

To generate adversarial texts, we first create the semantically consistent substitute word sets, which are then used to replace words in the original text, ensuring effective perturbations that increase the likelihood of deceiving VLP models. Previous methods typically employed the masked language model (MLM) [8] to generate substitutes for specific positions in the text. However, MLM predictions rely solely on context, which can lead to semantically inconsistent words. For example, in the sentence "I [MASK] you." [MASK] may be predicted as either "love" or "hate". This inconsistency can severely degrade the effectiveness of generating adversarial text.

To address this issue, we use the counter-fitting word vector [22] to generate the substitute word set. Specifically, given a clean text $t_i = [x_1, ..., x_j, ..., x_L]$, where $x_j$ denotes the $j$-th word in the sentence, we generate a textual adversarial example $t_i'$ by replacing words in $t_i$. For each word $x_j$ in the sentence, if its corresponding word vector $\mathbf{v}_{x_j}$ exists in the counter-fitting word vector set $\mathbf{V}_{cf}$, we select all synonyms $x_j'$ whose word vectors $\mathbf{v}_{x_j'} \in \mathbf{V}_{cf}$ have a cosine similarity greater than $\tau$ with $\mathbf{v}_{x_j}$. If $\mathbf{v}_{x_j}$ does not exist in $\mathbf{V}_{cf}$, we generate $k$ synonyms using the method from BERT-Attack [14]. This process can be formally defined as:

$$C(x_j) = \begin{cases} x_j' \mid \cos(\mathbf{v}_{x_j'}, \mathbf{v}_{x_j}) > \tau, & \text{if } \mathbf{v}_{x_j} \in \mathbf{V}_{cf}, \\ \arg\max_k f_{\text{mlm}}(x_j), & \text{otherwise}, \end{cases} \tag{3}$$

where $C(x_j)$ denotes the set of substitute words for $x_j$, and $f_{mlm}(\cdot)$ represents the masked language model used in BERT-Attack to generate synonyms.

#### 4.1.2 Generating Textual Adversarial Example

Intuitively, image-text pairs with lower similarity on the surrogate model tend to have lower similarity on the victim model as well, making these adversarial image-text pairs more likely to succeed in

attacking the victim model. Therefore, to improve the attack success rate, we need to generate a textual adversarial example $t'_i$ by applying synonym replacement, such that its similarity with the corresponding image $v_i$ is lowest in the surrogate model. In several vision-language tasks, the text often consists of a limited number of tokens. Consequently, we restrict the replacement to a single word within the text to generate $t'_i$. Specifically, for each word $x_j$ in the text $t_i = [x_1, ..., x_j, ..., x_L]$, we iteratively substitute $x_j$ with each synonym from its substitute word set $C(x_j)$, thereby generating a set of candidate texts $\mathbb{C}(t_i)_j$. We then aggregate all candidate sets to form the comprehensive adversarial text collection $\mathbb{C}(t_i)$. The adversarial example $t'_i$ is subsequently selected based on the minimum cosine similarity to the original image, as formalized by the following equation:

$$t'_i = \underset{t_i^* \in \mathbb{C}(t_i)}{\operatorname{argmax}} - cos(E_T(t_i^*, f_\phi), E_I(v_i, f_\phi)), \tag{4}$$

where $cos(\cdot, \cdot)$ is the cosine similarity function, $E_T(t_i^*, f_\phi)$ denotes the text feature of $t_i^*$ extracted by the text encoder of the surrogate model $f_\phi$, $E_I(v_i, f_\phi)$ denotes the image feature of $v_i$ extracted by the image encoder of the surrogate model $f_\phi$.

## 4.2 Image Attack

### 4.2.1 Layer-Importance Based Initial Image Adversarial Example Generation

This step aims to generate an initial image adversarial example by minimizing its similarity to the original image. Existing methods [34, 35] extract layer-wise representations and minimize their similarity iteratively. However, they wrongly assume equal layer contributions, while actual influence varies. To address this, we propose a layer-importance based method to generate an initial image adversarial example. This method involves two steps: determining layer importance and generating an initial image adversarial example.

**Determining layer importance.** We conduct an experiment to quantify the contribution of each layer in the model. As illustrated in Figure 3, we present two similarity variation curves: (1) the cosine similarity between the output feature of an image $v_i$ under normal propagation, denoted as $E_I(v_i, f_\phi)$, and the output feature obtained after skipping the $l$-th layer, denoted as $E_{I \backslash l}(v_i, f_\phi)$; (2) the cosine similarity between the [CLS] token embedding at the $l$-th layer, $E_I(v_i, f_\phi)_{l,1}$, and the [CLS] token embedding at the top layer $L_p$, $E_I(v_i, f_\phi)_{L_p,1}$. Further experimental details are provided in Appendix B. The results indicate that as the layer index increases, skipping a layer exerts a more pronounced influence on the final output, highlighting the greater significance of higher layers. Similarly, the similarity trend of the [CLS] token follows a comparable pattern, exhibiting

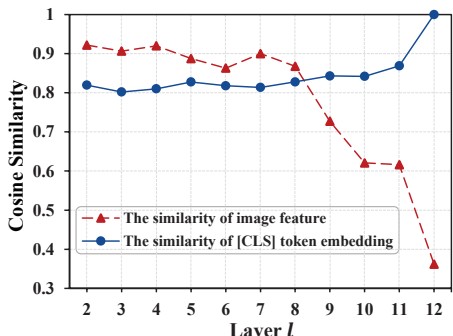

Figure 3: The cosine similarity of image feature and [CLS] token embedding across Layers.

a smoother variation. Since different training strategies result in varying parameter sensitivities, more substantial changes are observed in the top layers. Assigning higher importance weights to higher layers can lead to adversarial examples that overfit the surrogate model, thereby reducing their effectiveness across different models. The smooth variation observed in the [CLS] token suggests that this issue can be mitigated by narrowing the importance weight gap between lower and higher layers. Consequently, employing the cosine similarity between $E_I(v_i, f_\phi)_{l,1}$ and $E_I(v_i, f_\phi)_{L_p,1}$ as the importance weight is a reasonable choice. Thus, the $l$-th layer importance weight is defined as:

$$w_{i,l} = cos(E_I(v_i, f_\phi)_{l,1}, E_I(v_i, f_\phi)_{L_p,1}). \tag{5}$$

In Eq. (5), a higher value of $w_{i,l}$ indicates greater layer importance in generating adversarial examples.

**Generate initial image adversarial example.** Given the computed layer importance, we optimize an adversarial image that differs significantly from the original in important layers by minimizing the weighted sum of feature similarities. Specifically, for an original image $v_i$, we introduce a random perturbation $\delta \sim U(-\epsilon, \epsilon)$ to obtain $v'_i = v_i + \delta$. We then compute the cosine similarity between the features $E_I(v_i, f_\phi)_{l,j}$ and $E_I(v'_i, f_\phi)_{l,j}$ extracted from the $j$-th token at the $l$-th layer of the surrogate

model for both $v_i$ and $v_i'$. This optimization is formulated as:

$$\mathcal{L}_l = \sum_{l=1}^{L_p} w_{i,l} \times \frac{1}{D_p} \sum_{j=1}^{D_p} \cos(E_I(v_i, f_\phi)_{l,j}, E_I(v_i', f_\phi)_{l,j}), \tag{6}$$

where $D_p$ denotes the number of tokens per layer. Minimizing $\mathcal{L}_l$ reduces the similarity between the adversarial image $v_i'$ and the original image $v_i$. Finally, we apply PGD [21] to optimize this objective and generate the initial image adversarial example $v_i'$.

### 4.2.2 Contrastive Learning Based Image Adversarial Example Optimization

To further reduce the similarity between positive image-text pairs and increase the similarity between negative image-text pairs—thereby encouraging both the surrogate and victim models to retrieve unmatched texts for $v_i'$ and ultimately enhancing the attack success rate—we employ a contrastive learning approach to optimize the adversarial image $v_i'$.

Specifically, let $T_p$ denote the set of textual adversarial examples generated from the texts matched with $v_i'$ in a batch of image-text pairs, along with their corresponding original texts. Let $T_n$ denote the set of adversarial texts that are unmatched with $v_i'$ within the same batch. During the optimization process, we design the following loss function:

$$\mathcal{L}_c = \sum_{v_i^* \in \text{Trans}(v_i')} (\lambda \sum_{t_i' \in T_p} \cos(E_T(t_i', f_\phi), E_I(v_i^*, f_\phi)) + \sum_{t_j' \in T_n} \cos(E_T(t_j', f_\phi), E_I(v_i^*, f_\phi)), \tag{7}$$

where $\text{Trans}(v_i')$ is the scale transformation function, and $\lambda$ is the penalty factor for positive image-text pairs in contrastive learning.

By means of this loss function, on the surrogate model $f_\phi$, for positive image-text pairs, we minimize their similarity, that is, increase their distance in the feature space; for negative image-text pairs, we maximize the similarity of their feature vectors, that is, reduce the distance between the two in the feature space. As a result, the optimized adversarial image example is more likely to retrieve negative texts, thereby improving the attack success rate. Finally, we utilize the PGD to optimize $\mathcal{L}_c$ and obtain the refined image adversarial example $v_i'$.

### 4.3 The Overall Procedure

HQA-VLAttack begins by extracting a batch of image-text pairs from the datasets $D_t$ and $D_v$ in each round and optimizing them through the following procedure to generate $D_t'$ and $D_v'$. The process starts with a text attack: given an original image-text pair $(v_i, t_i)$, HQA-VLAttack first determines the substitute word set for each candidate word in $t_i$. Using these substitute sets, it constructs the textual adversarial example $t_i'$. Following this, an iterative image attack is conducted. In each iteration, HQA-VLAttack first initializes the image adversarial example $v_i'$, then uses a contrastive learning-based method to further optimize the image adversarial example. The detailed algorithm procedure of HQA-VLAttack is given in Appendix C.

## 5 Experiment

### 5.1 Experimental Settings

**Dataset.** We conduct experiments on three widely-used public multimodal datasets Flickr30K [25], MSCOCO [16], and RefCOCO+ [36]. The detailed dataset description is shown in Appendix D. For the image-text retrieval task, we conduct experiments on the Flickr30K test set, which contains 1,000 images and 5,000 captions, as well as on the MSCOCO test set, which includes 5,000 images and approximately 25,000 captions. For visual grounding and image captioning tasks, we use 3,000 images and 15,000 captions from RefCOCO+ as well as 10,000 images and 50,000 captions from MSCOCO, respectively. We adopt the Karpathy split for experimental evaluation.

**Models.** We follow [19, 10] to evaluate two popular VLP architectures, the fused VLP and aligned VLP models. Specifically, we select ALBEF [13] and TCL [33] as representatives of the fused VLP category. ALBEF integrates a 12-layer visual transformer (ViT-B/16) [9] as the image encoder, and

Table 1: **Attack success rate (%) in image-text retrieval on the Flickr30K dataset.** We present the attack success rate metric R@1 for both IR and TR tasks, indicating the success of attacks at Rank 1. * indicates white-box attacks.

| Surrogate Model | Victim Model | ALBEF | | TCL | | CLIP$_{\text{ViT}}$ | | CLIP$_{\text{CNN}}$ | |
|---|---|---|---|---|---|---|---|---|---|
| | Attack Method | TR R@1 | IR R@1 | TR R@1 | IR R@1 | TR R@1 | IR R@1 | TR R@1 | IR R@1 |
| ALBEF | PGD | 52.45* | 58.65* | 3.06 | 6.79 | 8.96 | 13.21 | 10.34 | 14.65 |
| | BERT-Attack | 11.57* | 27.46* | 12.64 | 28.07 | 29.33 | 43.17 | 32.69 | 46.11 |
| | Sep-Attack | 65.69* | 73.95* | 17.60 | 32.95 | 31.17 | 45.23 | 32.83 | 45.49 |
| | Co-Attack | 77.16* | 83.86* | 15.21 | 29.49 | 23.60 | 36.48 | 25.12 | 38.89 |
| | SGA | 97.24* | 97.28* | 45.42 | 55.25 | 33.38 | 44.16 | 34.93 | 46.57 |
| | DRA | 96.14* | 96.63* | 49.74 | 58.83 | 39.14 | 48.39 | 41.38 | 51.66 |
| | **HQA-VLAttack** | **99.79*** | **99.98*** | **73.02** | **77.60** | **52.15** | **62.05** | **59.64** | **65.59** |
| TCL | PGD | 6.15 | 10.78 | 77.87* | 79.48* | 7.48 | 13.72 | 10.34 | 15.33 |
| | BERT-Attack | 11.89 | 26.82 | 14.54* | 29.17* | 29.69 | 44.49 | 33.46 | 46.06 |
| | Sep-Attack | 20.13 | 36.48 | 84.72* | 86.07* | 31.29 | 44.65 | 33.33 | 45.80 |
| | Co-Attack | 23.15 | 40.04 | 77.94* | 85.59* | 27.85 | 41.19 | 30.74 | 44.11 |
| | SGA | 48.91 | 60.34 | 98.37* | 98.81* | 33.87 | 44.88 | 37.74 | 48.30 |
| | DRA | 51.09 | 61.79 | 98.21* | 98.33* | 40.25 | 48.94 | 42.91 | 52.49 |
| | **HQA-VLAttack** | **62.88** | **71.70** | **99.79*** | **99.93*** | **52.39** | **59.41** | **55.43** | **62.44** |
| CLIP$_{\text{ViT}}$ | PGD | 2.50 | 4.93 | 4.85 | 8.17 | 70.92* | 78.61* | 5.36 | 8.44 |
| | BERT-Attack | 9.59 | 22.64 | 11.80 | 25.07 | 28.34* | 39.08* | 30.40 | 37.43 |
| | Sep-Attack | 9.59 | 23.25 | 11.38 | 25.60 | 79.75* | 86.79* | 30.78 | 39.76 |
| | Co-Attack | 10.57 | 24.33 | 11.94 | 26.69 | 93.25* | 95.86* | 32.52 | 41.82 |
| | SGA | 13.40 | 27.22 | 16.23 | 30.76 | 99.08* | 98.94* | 38.76 | 47.79 |
| | DRA | 12.51 | 30.00 | 14.65 | 30.62 | 98.77* | 99.00* | 45.47 | 50.74 |
| | **HQA-VLAttack** | **25.13** | **41.98** | **24.66** | **44.00** | **100.00*** | **100.00*** | **74.07** | **77.19** |
| CLIP$_{\text{CNN}}$ | PGD | 2.09 | 4.82 | 4.00 | 7.81 | 1.10 | 6.60 | 86.46* | 92.25* |
| | BERT-Attack | 8.86 | 23.27 | 12.33 | 25.48 | 27.12 | 37.44 | 30.40* | 40.10* |
| | Sep-Attack | 8.55 | 23.41 | 12.64 | 26.12 | 28.34 | 39.43 | 91.44* | 95.44* |
| | Co-Attack | 8.79 | 23.74 | 13.10 | 26.07 | 28.79 | 40.03 | 94.76* | 96.89* |
| | SGA | 11.42 | 24.80 | 14.91 | 28.82 | 31.24 | 42.12 | 99.24* | 99.49* |
| | DRA | 12.20 | 26.59 | 14.33 | 29.29 | 35.21 | 45.94 | 99.11* | 99.49* |
| | **HQA-VLAttack** | **20.75** | **38.66** | **22.13** | **42.45** | **62.82** | **69.46** | **99.87*** | **100.00*** |

employs two 6-layer transformers as the text encoder and multimodal encoder, respectively. TCL shares the same architectural framework as ALBEF but is distinguished by its unique pre-training objectives. For the aligned VLP model, we focus on CLIP [26], which offers two distinct image encoder variants: CLIP$_{\text{ViT}}$ and CLIP$_{\text{CNN}}$. These variants leverage ViT-B/16 and ResNet-101 [12] as their respective base architectures for the image encoder.

**Baselines.** We compare HQA-VLAttack with the following baselines: (1) **PGD** [21] is a white-box image adversarial attack method that iteratively maximizes model loss under perturbation constraints via projected gradient descent. (2) **BERT-Attack** [14] is a black-box query-based textual adversarial attack method that crafts context-aware substitutions via BERT to fool NLP models with minimal edits. (3) **Sep-Attack** [19] is a black-box transfer-based multimodal adversarial attack method that separately perturbs unimodal data without any cross-modal interactions. (4) **Co-Attack** [37] is a white-box multimodal adversarial attack method that collaboratively perturbs both image and text modalities to enhance adversarial effects. (5) **SGA** [19] is a black-box transfer-based multimodal adversarial attack method that uses set-level attacks to boost adversarial transferability in vision-language models. (6) **DRA** [10] is a recent black-box transfer-based multimodal adversarial attack method that enhances adversarial transferability by diversifying adversarial examples along the intersection region of the adversarial trajectory.

**Evaluation Metrics.** We use the Attack Success Rate (ASR) as the primary evaluation metric to assess the transferability of adversarial attacks in both white-box and black-box settings. The ASR reflects the overall success rate of the attacks, with a higher ASR indicating a higher quality attack method. Additionally, we employ IR R@k, which measures the proportion of cases where none of the top-k image retrieval results contain the correct image. Similarly, we use TR R@k to represent the proportion of instances where none of the top-k caption retrieval results include the correct matching caption.

**Implementation Details.** In our experiments, we adopt adversarial attack settings of SGA in order to ensure the fairness of the comparison. For image attacks, we employ PGD with perturbation bound $\epsilon_v = 2/255$, step size $\alpha = 0.5/255$, and iteration steps $N = 10$. We leverage a combination of BERT-Attack and counter-filter word vectors to craft adversarial texts. The perturbation boundary is set to $\epsilon_t = 1$. For BERT-Attack, the length of the word list is $W = 10$. For the word vectors, the similarity threshold is set to $\tau = 0.4$. In contrastive learning, the positive pair penalty factor $\lambda$ is set

Table 2: **Attack success rate (%) in image-text retrieval on the MSCOCO Dataset.** We present the attack success rate metric R@1 for both IR and TR tasks, indicating the success of attacks at Rank 1. * indicates white-box attacks.

| Surrogate Model | Victim Model | ALBEF | | TCL | | CLIP$_{ViT}$ | | CLIP$_{CNN}$ | |
|---|---|---|---|---|---|---|---|---|---|
| | Attack Method | TR R@1 | IR R@1 | TR R@1 | IR R@1 | TR R@1 | IR R@1 | TR R@1 | IR R@1 |
| **ALBEF** | PGD | 76.70* | 86.30* | 12.46 | 17.77 | 13.96 | 23.10 | 17.45 | 23.54 |
| | BERT-Attack | 24.39* | 36.13* | 24.34 | 33.39 | 44.94 | 52.28 | 47.73 | 54.75 |
| | Sep-Attack | 82.60* | 89.88* | 32.83 | 42.92 | 44.03 | 54.46 | 46.96 | 55.88 |
| | Co-Attack | 79.87* | 87.83* | 32.62 | 43.09 | 44.89 | 54.75 | 47.30 | 55.64 |
| | SGA | 96.75* | 96.95* | 58.56 | 65.38 | 57.06 | 65.25 | 58.95 | 66.52 |
| | DRA | 96.57* | 96.47* | 60.69 | 67.46 | 61.69 | 67.43 | 62.32 | 69.22 |
| | **HQA-VLAttack** | **99.97*** | **99.97*** | **85.69** | **87.81** | **75.35** | **80.20** | **77.52** | **81.64** |
| **TCL** | PGD | 10.83 | 16.52 | 59.58* | 69.53* | 14.23 | 22.28 | 17.25 | 23.12 |
| | BERT-Attack | 35.32 | 45.92 | 38.54* | 48.48* | 51.09 | 58.80 | 52.23 | 61.26 |
| | Sep-Attack | 41.71 | 52.97 | 70.32* | 78.97* | 50.74 | 60.13 | 51.90 | 61.26 |
| | Co-Attack | 46.08 | 57.09 | 85.38* | 91.39* | 51.62 | 60.46 | 52.13 | 62.49 |
| | SGA | 65.93 | 73.30 | 98.97* | 99.15* | 56.34 | 63.99 | 59.44 | 65.70 |
| | DRA | 68.06 | 75.86 | 98.99* | 99.15* | 63.30 | 63.99 | 64.24 | 65.70 |
| | **HQA-VLAttack** | **81.90** | **86.32** | **99.97*** | **99.97*** | **70.77** | **76.22** | **73.52** | **78.22** |
| **CLIP$_{ViT}$** | PGD | 7.24 | 10.75 | 10.19 | 13.74 | 54.79* | 66.85* | 7.32 | 11.34 |
| | BERT-Attack | 20.34 | 29.74 | 21.08 | 29.61 | 45.06* | 51.68* | 44.54 | 55.32 |
| | Sep-Attack | 23.41 | 34.61 | 25.77 | 36.84 | 68.52* | 77.94* | 43.11 | 49.76 |
| | Co-Attack | 30.28 | 42.67 | 32.84 | 44.69 | 97.98* | 98.80* | 55.08 | 62.51 |
| | SGA | 33.41 | 44.64 | 37.54 | 47.76 | 99.79* | 99.79* | 58.93 | 65.83 |
| | DRA | 35.96 | 48.00 | 36.32 | 48.56 | 99.66* | 99.70* | 64.41 | 69.99 |
| | **HQA-VLAttack** | **56.05** | **66.03** | **54.10** | **65.25** | **100.00*** | **100.00*** | **88.93** | **89.26** |
| **CLIP$_{CNN}$** | PGD | 7.01 | 10.62 | 10.08 | 13.65 | 4.88 | 10.70 | 76.99* | 84.20* |
| | BERT-Attack | 23.38 | 34.64 | 24.58 | 29.61 | 51.28 | 57.49 | 54.43* | 62.17* |
| | Sep-Attack | 26.53 | 39.29 | 30.26 | 41.51 | 50.44 | 57.11 | 88.72* | 92.49* |
| | Co-Attack | 29.83 | 41.97 | 32.97 | 43.72 | 53.10 | 58.90 | 96.72* | 98.56* |
| | SGA | 31.61 | 43.00 | 34.81 | 45.95 | 56.62 | 60.77 | 99.61* | 99.80* |
| | DRA | 33.26 | 45.15 | 33.89 | 46.49 | 59.60 | 64.87 | 99.51* | 99.70* |
| | **HQA-VLAttack** | **52.20** | **62.13** | **51.11** | **62.49** | **82.72** | **84.56** | **100.00*** | **100.00*** |

Table 3: **Cross-Task Transferability.** The Baseline represents the original performance of IC and VG on clean data. We utilize ALBEF to generate multi-modal adversarial examples for attacking both Visual Grounding (VG) and Image Captioning (IC).

| Attack | ITR → VG | | | ITR → IC | | | | |
|---|---|---|---|---|---|---|---|---|
| | Val ↓ | TestA ↓ | TestB ↓ | B@4 ↓ | METEOR ↓ | ROUGE-L ↓ | CIDEr ↓ | SPICE ↓ |
| Baseline | 58.46 | 65.89 | 46.25 | 39.7 | 31.0 | 60.0 | 133.3 | 23.8 |
| Co-Attack | 54.26 | 61.80 | 43.81 | 37.4 | 29.8 | 58.4 | 125.5 | 22.8 |
| SGA | 53.55 | 61.19 | 43.71 | 34.8 | 28.4 | 56.3 | 116.0 | 21.4 |
| DRA | 53.88 | 61.18 | 43.38 | 34.8 | 28.4 | 56.4 | 115.9 | 21.4 |
| **HQA-VLAttack** | **46.48** | **54.31** | **36.90** | **31.8** | **26.8** | **54.1** | **104.6** | **19.8** |

to $-10$, batch size is set to 16. Image scale sets $S = \{0.50, 0.75, 1.00, 1.25, 1.50\}$. Similarly, the caption set is enlarged by augmenting the most matching caption pairs for each image in the dataset, with the size of approximately five.

## 5.2 Experimental Results

### 5.2.1 Image-Text Retrieval Comparison

Our experiments focus on the Image-Text Retrieval (ITR) task, where we generate adversarial examples across various surrogate models and evaluate the effectiveness of our method using the Attack Success Rate (ASR) for both white-box and transfer attacks. As shown in Table 1 and Table 2, our approach outperforms state-of-the-art methods in ASR on both Flickr30K and MSCOCO datasets. Specifically, our method achieves nearly 100% ASR in white-box attacks, with both Text Retrieval (TR) and Image Retrieval (IR) ASR reaching 100% when attacking CLIP$_{ViT}$ on both Flickr30K and MSCOCO.

In black-box attacks using models with the same architecture, attacking TCL with ALBEF as the surrogate model results in ASR increases of 23.28% (TR) and 18.77% (IR) on Flickr30K, as well as 25.00% (TR) and 20.35% (IR) on MSCOCO. Similarly, when CLIP$_{ViT}$ is used as the surrogate model to attack CLIP$_{CNN}$, ASR gains of 28.60% (TR) and 26.45% (IR) on Flickr30K, with corresponding gains of 24.52% (TR) and 19.27% (IR) on MSCOCO.

For attacks involving different model architectures, using ALBEF as the surrogate model to attack $CLIP_{ViT}$ results in ASR improvements of 13.01% (TR) and 13.66% (IR) on Flickr30K, as well as 13.66% (TR) and 12.77% (IR) on MSCOCO. Conversely, using $CLIP_{ViT}$ as the surrogate model to attack ALBEF yields ASR enhancements of 12.62% (TR) and 11.98% (IR) on Flickr30K, with corresponding enhancements of 20.09% (TR) and 18.03% (IR) on MSCOCO.

These results demonstrate that HQA-VLAttack is a high quality attack method, significantly outperforming other approaches in terms of ASR.

### 5.2.2 Cross-Task ASR Comparison

To verify that HQA-VLAttack is not only effective in Image-Text Retrieval but also in other tasks, we conduct experiments on Image Captioning and Visual Grounding. These tasks demand strong cross-modal interaction and alignments, which are core components of multimodal learning.

**Image Captioning.** Image captioning is a generative task where the model [30, 29, 1] first encodes the input image and then generates the corresponding textual description using a decoder based on the encoded image features. In our experiment, we select ALBEF as the surrogate model to generate adversarial examples for the Image-Text Retrieval (ITR) task and use BLIP as the victim model for image captioning. The experiment is conducted on the MSCOCO dataset, and the generated captions are evaluated using the following metrics: BLEU-4 (B@4) [24], METEOR [3], ROUGE [15], CIDEr [28], and SPICE [2]. The results are shown in the Table 3. It can be seen that compared with the second-best results, HQA-VLAttack improves the BLEU score by up to 3.0% and the CIDEr score by up to 11.3%.

**Visual Grounding.** The task of visual grounding aims to locate the region in the image that corresponds to a specific textual description. In our experiment, we use ALBEF as the surrogate model to generate adversarial examples for the Image-Text Retrieval (ITR) task and select the ALBEF model fine-tuned for visual grounding as the victim model. The experiment is conducted using the RefCOCO+ dataset, and we employ Val, TestA, and TestB as evaluation metrics. As shown in Table 3, HQA-VLAttack outperforms other methods significantly.

We further conduct experiments to assess the adversarial transferability of our method on Multimodal Large Language Models (MLLMs)[23]. Figure 5 shows adversarial examples generated by our method that successfully mislead state-of-the-art closed-source MLLMs to produce incorrect responses. Detailed experimental settings as well as additional experiments are provided in Appendix G.

### 5.2.3 Ablation Study and Parameter Investigation

We conduct experiments on ITR to evaluate the effects of different modules and batch sizes $B$ in the proposed HQA-VLAttack. Specifically, we use the Flickr30K dataset, with ALBEF as the surrogate model, and employ the Attack Success Rate (ASR, %) as the metric. We also investigate the impact of the penalty factor $\lambda$ in Appendix F.

**Module Ablation Experiment.** When investigating the effectiveness of different components, we select TCL as the victim model. The results are shown in the Figure 4a. "w/o CF" refers to the omission of the counter-fitting word vector for generating substitute words during the determining the substitute word set phase. "w/o LI" refers to the case where no layer importance is used during the layer-importance based initial image adversarial example generation phase, and each $w_{i,l}$ is set to 1. "w/o IG" refers to the omission of the Layer-Importance Based Initial Image Adversarial Example Generation phase in the

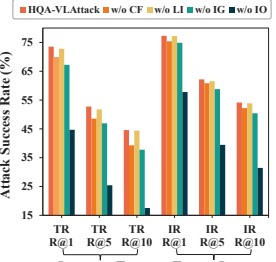 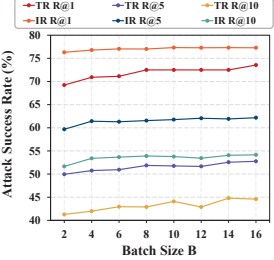

(a) Component Ablation Analysis    (b) Impact of Batch Size $B$ on ASR

Figure 4: Ablation Study on Component Effectiveness and Batch Size Impact on Attack Success Rate.

image attack process. "w/o IO" refers to the omission of the Contrastive Learning Based Image Ad-

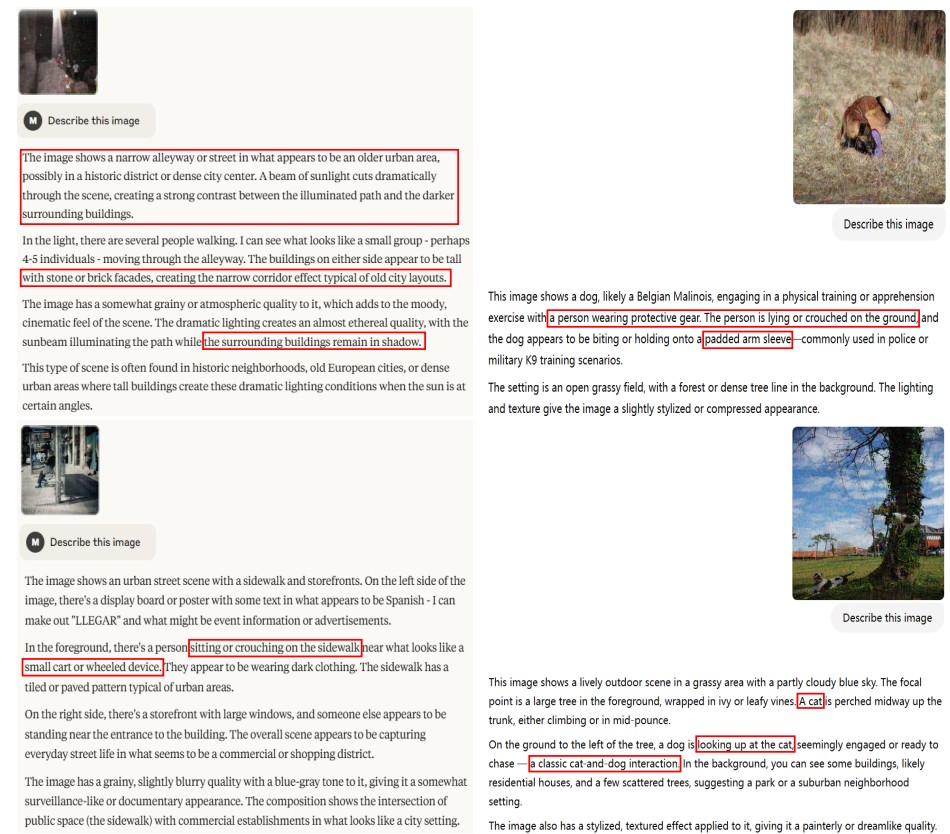

Figure 5: Adversarial Transferability between GPT-4o and Claude-3.7 Sonnet. The images on the left show the responses generated by Claude-3.7 Sonnet when provided with adversarial images and the prompt "Describe this image", while the images on the right display the outputs produced by GPT-4o under the same prompt.

versarial Example Optimization. It is evident that all components contribute to the HQA-VLAttack's performance, validating the effectiveness of each proposed component.

**Batch Size** $B$**.** To investigate the impact of different batch sizes $B$ on attack success rate, we conduct an ablation analysis, and the results are shown in Figure 4b. As $B$ increases, the contrastive learning process incorporates more unmatched texts, which helps guide the generation of adversarial images, leading to a noticeable increase in the attack success rate. These results highlight the effectiveness of a larger batch size in improving adversarial transferability. However, to balance attack performance with computational efficiency, we ultimately selected $B = 16$ as the optimal batch size.

## 6 Conclusion

In this paper, we propose a novel method, HQA-VLAttack, to achieve high-quality adversarial attacks against Vision-Language Pre-training (VLP) models. For text attack, HQA-VLAttack ensures semantic consistency between the substitute word and the original word by utilizing counter-fitting word vectors to identify an appropriate set of substitute words. For image attack, HQA-VLAttack generates the initial image using a layer-importance-based approach, minimizing the similarity between the initial and original images. Moreover, by contrastive learning-based optimization, HQA-VLAttack reduces the similarity between positive image-text pairs while enhancing the similarity between negative image-text pairs. This forces the victim model to prioritize the retrieval of negative image-text pairs. Extensive experimental results demonstrate that HQA-VLAttack significantly outperforms strong baselines without requiring queries to the victim model, underscoring its effectiveness as a high-quality attack. In future work, we aim to explore further optimization strategies to refine the model and enhance its adversarial attack performance.

## Acknowledgments and Disclosure of Funding

This work was supported by National Natural Science Foundation of China (No. 62206038, 62106035), Liaoning Binhai Laboratory Project (No. LBLF-2023-01), Chunhui Project Foundation of the Education Department of China (No. HZKY20220419), and Xiaomi Young Talents Program.

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
