# OpenReview forum: "HQA-VLAttack: Towards High Quality Adversarial Attack on Vision-Language Pre-Trained Models"
_NeurIPS.cc/2025/Conference — NeurIPS 2025 poster_

### Official Review · Reviewer_NhC3 · 2025-06-03

**Clarity:** 2
**Significance:** 2
**Originality:** 3
**Rating:** 4
**Confidence:** 4

**Summary:**

This paper proposes HQA-VLAttack, a two-stage adversarial attack framework for Vision-Language Pre-trained (VLP) models. Text attacks employ counter-fitting word vectors for semantically consistent substitutions. Image attacks combine layer-importance guided initialization  with contrastive learning optimization. A core aim is to decrease positive image-text pair similarity while increasing negative image-text pair similarity.

**Questions:**

- Can the authors explicitly define how "negative pairs" in Figure 1 are constructed? Are they perturbed original negative pairs, or adversarial versions of negative pairs derived from attacks on positive counterparts
- The principle behind the text attack strategy in Section 4.1: first collecting semantically similar words then selecting the one that minimizes image-text similarity, lacks sufficient explanation regarding its balance of semantics and attack efficacy. Can the authors detail why in Section 4.1, semantically similar words are first identified (4.1.1), and then the final attack selects the word minimizing image-text cosine similarity (4.1.2)? Is this to balance semantic plausibility and attack strength, and how is this balance systematically managed?
- Considering rapid VLM advancements, would the authors conduct and present quantitative evaluations of HQA-VLAttack on more recent SOTA models (e.g., Qwen, LLaVA) and diverse tasks like VQA? Like Table 3.

**Ethical Concerns:**

["NO or VERY MINOR ethics concerns only"]

**Final Justification:**

Overall, the paper presents an attack method targeting VLMs. While the novelty and impact of the work are somewhat limited, the authors are able to justify their approach in a reasonable and coherent manner. As such, I consider a borderline accept rating to be a fair and balanced evaluation.

**Limitations:**

yes

**Quality:**

3

**Strengths And Weaknesses:**

Strength:

- The strategy of manipulating negative image-text pair similarity via contrastive learning is novel and potentially addresses a gap in prior research.
- Text attacks aim for semantic consistency using counter-fitting word vectors.
- Image attacks utilize a layer-importance based perturbation initialization strategy.
- The paper demonstrates high attack success rates on the tested VLP models and datasets.

Weaknesses:
-  The motivation presented in Figure 1 regarding "negative pairs" is ambiguous and requires significant clarification. It is unclear whether these negative samples are: (a) clean, originally negative (i.e., non-matching) image-text pairs that are then subjected to some form of perturbation for the purpose of this illustrative plot; or (b) if they are adversarial versions of originally negative image-text pairs, where the perturbation strategy applied to them is somehow derived from an attack on corresponding positive pairs
- The experimental validation, while covering several established datasets (Flickr30K, MSCOCO, RefCOCO+) and older VLP architectures (ALBEF, TCL, CLIP), does not sufficiently address the current, rapidly evolving landscape of Vision-Language Models. To establish the "high quality" and broad applicability of HQA-VLAttack in the current research context, validation on more recent and powerful SOTA models (e.g., Qwen, LLaVA) is crucial. Furthermore, the range of tasks could be expanded, for instance, with more complex VQA scenarios. While Appendix G mentions experiments on MLLMs with a reference to GPT-4 and visualizations or brief mentions in an appendix are not a substitute for comprehensive quantitative analysis on these advanced systems.

---

> ### Author Rebuttal · Authors · 2025-07-31
>
> Thanks very much for your insightful and positive comments.
>
> **1. Clarification about "negative pairs" in Figure 1.**
>
> By "negative pairs", we mean the non-matching image-text pairs. For all the mentioned methods SGA, DRA and HQA-VLAttack in Figure 1, the "negative pairs" are constructed by the perturbed images and non-matching perturbed texts drawn from other examples in the same batch.
> The similarity of "negative pairs" is averaged over all adversarial images, where the similarity of each adversarial image is averaged over five non-matching perturbed texts.
>
> Figure 1 aims to show the difference between our method and previous works. Specifically, SGA and DRA mainly focus on reducing the similarity of positive pairs, but they unintentionally suppress the similarity of negative pairs as well, which can limit the attack performance. In contrast, our method explicitly encourages higher similarity between perturbed images and non-matching perturbed texts, thus increasing the chance of irrelevant retrievals and improving the attack performance.
>
> We will add more explanations about "negative pairs" in the final version.
>
> **2. Clarification about the text attack strategy in Section 4.1.**
>
> The text attack strategy aims to generate high-quality textual adversarial examples, which should satisfy the semantic consistency principle while misleading the model. If the perturbed text loses the semantic consistency with the original text, it cannot serve as a valid adversarial example in vision-language tasks. Therefore, we first guarantee the semantic consistency before optimizing for the attack effectiveness.
>
> To achieve this, we design the following two-step strategy.
>
> **(1) Step 1 (Section 4.1.1).** In this step, we collect semantically similar words by retrieving counter-fitted synonyms for each word in the text. This ensures that the perturbed text has the semantic consistency with the original text. In contrast, previous methods like DRA and SGA use MLM-based substitutions, which probably introduce words that are semantically inconsistent. For ease of understanding, we list some generated examples that our method (using the counter-fitting based strategy) and previous methods (using the MLM-based strategy) are as follows.
>
>  | Original Caption | Previous Strategy | Our Strategy |
>  | --- | --- | --- |
>  | Road workers paving a road | Road young paving a road | Road officer paving a road |
>  | Students work together in groups | Online work together in groups | Kiddies work together in groups |
>  | A happy family at thanksgiving | A happy future at thanksgiving | A happy family at festivals |
>  | A car parked at the beach | A car parked at the each | A car parked at the waikiki |
>
> From these examples, we can find that the strategy used in Section 4.1.1 is reasonable and effective.
>
> **(2) Step 2 (Section 4.1.2).** In this step, we select the word that minimizes the image-text cosine similarity on the surrogate model from the semantically similar candidates. By choosing the word with the most similarity reduction, we can diminish the surrogate model's confidence and increase the attack success rate.
>
> In summary, Step 1 aims to keep the adversarial text semantically consistent and plausible, and Step 2 aims to improve the attack performance. We will add more explanations in the final version.
>
> **3. Evaluation on recent SOTA MLLMs.**
>
> We have conducted experiments on two recent SOTA MLLMs: Qwen2.5-VL (`Qwen2.5-VL-32B-Instruct`) and LLaVA-Next (`llava-v1.6-34b-hf`) under the image captioning task and visual question answering task. Specifically, we use ALBEF as the surrogate model to generate adversarial examples, with a perturbation budget of 16/255, 80 PGD steps, and a step size of 0.5/255.
>
> For the image captioning task, we randomly sample 128 image-text pairs from the Flickr30K dataset and generate adversarial images based on these clean pairs. Then we feed these adversarial images into Qwen2.5-VL and LLaVA-Next to generate captions. The generated captions are evaluated with the widely-used metrics BLEU-4, METEOR, ROUGE-L, CIDEr, and SPICE, where lower values indicate better attack performance.
>
> For the visual question answering task, we randomly sample 128 examples from the VQAv2 dataset and apply perturbations to both the input images and their corresponding questions. The adversarial images and adversarial questions are then jointly fed into Qwen2.5-VL and LLaVA-Next to generate answers. The generated answers are evaluated using the widely-used metric accuracy, where a lower accuracy indicates better attack performance. The detailed results are as follows.
>
> Note: The **Original** represents the performance on clean data.
>
> * **Image Captioning Results.**
>
> \> Attack Qwen2.5-VL.
>
>  | Method | BLEU-4  | METEOR  | ROUGE-L  | CIDEr  | SPICE  |
>  | --- | --- | --- | --- | --- | --- |
>  | Original | 13.1 | 20.9 | 39.9 | 49.9 | 14.3 |
>  | SGA | 12.3 | 18.6 | 37.6 | 42.5 | 12.6 |
>  | DRA | 10.3 | 18.2 | 36.0 | 36.7 | 11.6 |
>  | **HQA-VLAttack** | **7.4** | **18.1** | **33.9** | **25.3** | **11.4** |
>
> \> Attack LLaVA-Next.
>
>  | Method | BLEU-4  | METEOR  | ROUGE-L  | CIDEr  | SPICE  |
>  | --- | --- | --- | --- | --- | --- |
>  | Original | 14.1 | 15.7 | 33.2 | 42.5 | 10.6 |
>  | SGA | 5.8 | 13.8 | 30.1 | 30.8 | 9.9 |
>  | DRA | 5.8 | 13.3 | 28.8 | 28.7 | 9.5 |
>  | **HQA-VLAttack** | **4.8** | **11.9** | **27.8** | **23.7** | **7.8** |
>
> * **Visual Question Answering Results.**
>
>  | Method | Accuracy  (Qwen2.5-VL) | Accuracy  (LLaVA-Next) |
>  | --- | --- | --- |
>  | Original | 59.64 | 75.78 |
>  | SGA | 45.31 | 46.35 |
>  | DRA | 47.40 | 50.00 |
>  | **HQA-VLAttack** | **39.58** | **41.93** |
>
> These results demonstrate that HQA-VLAttack consistently outperforms strong baselines such as SGA and DRA when attacking the recent SOTA MLLMs. This further confirms the superiority of our method in the realistic black-box attack scenarios. We will add all these results in the final version.

---

> > ### Comment · Reviewer_NhC3 · 2025-08-02
> >
> > Thank you for the authors’ response. Having reviewed the rebuttal, I believe that assigning a borderline accept score remains a fair evaluation of the work.

---

> > > ### Author Response · Authors · 2025-08-03
> > >
> > > Dear Reviewer,
> > >
> > > Thank you once again for your valuable time and effort. We deeply value your insightful and constructive comments.
> > >
> > > Best regards,
> > >
> > > Authors

---

### Official Review · Reviewer_PSrx · 2025-06-29

**Clarity:** 3
**Significance:** 2
**Originality:** 3
**Rating:** 4
**Confidence:** 4

**Summary:**

This paper investigates adversarial transferability in vision-language pre-trained models and proposes a new method named High Quality transfer-based Adversarial Vision-Language Attack (HQA-VLAttack). The method enhances both textual and visual adversarial perturbations to improve attack effectiveness. For the textual component, it employs counter-fitting word vectors to ensure semantic consistency when generating substitute words. For the visual component, it introduces a layer-importance weighted strategy to guide the generation of image adversarial examples. Experimental results on multiple benchmark datasets demonstrate that HQA-VLAttack outperforms existing baselines in terms of attack success rate.

**Questions:**

My current rating is primarily based on the observed performance improvements. While these gains are substantial, my main concerns lie in the novelty of the proposed approach. Rating based on authors can effectively address these questions. I may lower it if they do not respond or if their responses are insufficient.

- How does the text attack differ from simple text augmentation techniques?
- What is the novelty in applying input transformation? Additionally, the scale transformation function should be explained in more detail, as readers unfamiliar with this field may find it difficult to follow.
- Can baseline methods also incorporate input transformation and contrastive loss for a fairer comparison? If so, how much additional improvement can be attributed solely to the layer-wise importance score and the substitute word set, when compared to the baselines?
- What is the computational overhead of the proposed method compared to baseline approaches?

**Ethical Concerns:**

["NO or VERY MINOR ethics concerns only"]

**Final Justification:**

The authors have addressed my questions, and I believe that maintaining my initial assessment remains a fair evaluation of the work. I encourage the authors to incorporate the clarifications provided in the rebuttal into the revised submission to improve clarity and completeness.

**Limitations:**

Yes

**Paper Formatting Concerns:**

I believe the appendix should be submitted with the main paper, not as a separate file. The code repository should be apporitely anonmouys, e.g., using https://anonymous.4open.science. Currently, it is using a dummy github user, which could still be in violation of double-blind review.

**Quality:**

2

**Strengths And Weaknesses:**

**Strengths**

- The paper addresses the important and practical problem of improving cross-model adversarial transferability for vision-language pre-trained models. Given the increasing real-world deployment of VLMs/VLPs, particularly in black-box settings, this direction is timely and impactful for the evaluation of adversarial robustness.
- The proposed image attack strategy, motivated by layer-wise cosine similarity (Figure 3), is insightful and, to my knowledge, novel within this field. It provides a more principled way of identifying impactful layers for generating transferable perturbations.
- The contrastive learning objective is technically sound and effectively encourages the retrieval of negative examples, improving the attack success rate.
- The experimental setup adheres closely to existing literature, enabling fair and direct comparisons. This clarity highlights the improvements in transferability achieved by the proposed method over strong baselines.

**Weaknesses**

- The text attack component (Section 4.1.1) lacks novelty. It essentially performs word-level substitution using counter-fitting vectors, which resembles a form of semantic-preserving text augmentation rather than a fundamentally new attack strategy.
- The use of contrastive loss over augmented image-text pairs to improve transferability is also not entirely novel. Input transformation techniques [1] are well-known for boosting adversarial transferability, yet this line of related work is not adequately discussed in the related work section. Additionally, contrastive loss as an attack objective could benefit from deeper theoretical or empirical justification.
- The ablation study in Figure 4(a) shows that the major gains come from the input transformation and contrastive learning components. In contrast, the counter-fitting word vector module contributes relatively little to the overall improvement in transferability.
- The proposed framework includes multiple additional modules, which may increase the computational cost of generating adversarial examples. However, the paper does not report any empirical analysis or runtime comparison to quantify this cost relative to baseline methods.

[1] Xie, C., Zhang, Z., Zhou, Y., Bai, S., Wang, J., Ren, Z., & Yuille, A. L. Improving transferability of adversarial examples with input diversity. In CVPR, 2019.

---

A Minor suggestion (not part of the rating justification):
A line of prior work, as well as some concurrent studies, explores universal adversarial perturbations to improve cross-model transferability. It might be beneficial to discuss these works in the related work section, as they may offer valuable insights and complementary perspectives relevant to the proposed approach.

- Zhang, P.F., Huang, Z., & Bai, G. Universal adversarial perturbations for vision-language pre-trained models. In SIGIR, 2024
- Fang, H., Kong, J., Yu, W., Chen, B., Li, J., Wu, H., ... & Xu, K. (2024). One perturbation is enough: On generating universal adversarial perturbations against vision-language pre-training models. arXiv preprint arXiv:2406.05491.
- Xu, B., Dai, X., Tang, D., & Zhang, K. (2025). One Surrogate to Fool Them All: Universal, Transferable, and Targeted Adversarial Attacks with CLIP. arXiv preprint arXiv:2505.19840.
- Huang, H., Erfani, S., Li, Y., Ma, X., & Bailey, J. (2025). X-Transfer Attacks: Towards Super Transferable Adversarial Attacks on CLIP. arXiv preprint arXiv:2505.05528.

---

> ### Author Rebuttal · Authors · 2025-07-31
>
> Thanks very much for your insightful and positive comments.
>
> **1. About the text attack component and simple text augmentation techniques.**
>
> (1) We would like to clarify that the novelty can also come from a simple yet effective algorithm design that solves some specific problems overlooked by previous works. Previous MLM-based (Masked Language Model-based) methods (e.g., SGA, DRA) have the semantic drift issue, i.e., just using MLM-generated substitutions may diverge from the original word meaning, which may harm both the semantic consistency and the fluency of adversarial texts. Our method used in Section 4.1.1 can alleviate the issue effectively. To verify this, we conduct the preference study using GPT-4o. Specifically, we randomly sample 128 image-text pairs from the Flickr30K dataset and generate adversarial texts using SGA, DRA, and HQA-VLAttack. For each time, GPT‑4o is provided with the original text and the three generated adversarial texts, and asked to select the one with the best semantic consistency and fluency. The results are as follows.
>
> | Method | Times selected as Best |
> | --- | --- |
> | SGA | 17 |
> | DRA | 40 |
> | HQA-VLAttack | **71** |
>
> The results show that our method can generate effective adversarial texts with higher similarity consistency. We will add all the explanations in the final version.
>
> (2) For the simple text augmentation, it aims to introduce random variations to increase data diversity, which does not need to consider the adversarial requirement. For the text attack, it aims to generate satisfactory adversarial text that has high semantic similarity (human-imperceptible) with the original text and can fool models to produce wrong predictions simultaneously. Although their goals are different, some simple text augmentation techniques like random insertion/swap/deletion, back-translation, and synonym replacement can also be used in the initial procedure of text attack. We will clarify their differences in the final version.
>
> **2. About the input transformation.**
>
> (1) The input transformation aims to improve the generalization ability of the adversarial examples, and it has been used in previous baselines SGA and DRA, which ensures that the performance gains come from our method itself.
>
> (2) The details of the scale transformation function are as follows. For a given image $I \in \mathbb{R}^{C \times H \times W}$, we perform the following steps:
>
> - Scaling: Resize the image to $rH \times rW$ using predefined ratios $r \in$ {0.50, 0.75, 1.00, 1.25, 1.50}.
> - Noise injection: Gaussian noise $\mathcal{N}(0, \sigma^2)$ can be added before resizing. In our experiments, we set $\sigma = 0.05$.
> - Resizing: Each scaled image is resized back to the original resolution via bicubic interpolation.
> - Contrastive integration: The resulting set $\text{Trans}(v'_i)$ is used in the contrastive learning loss to promote feature invariance across scales.
>
> (3) We have added the ablation study for the input transformation and the contrastive learning. Specifically, we use ALBEF as the surrogate model on the Flickr30K dataset. Adversarial examples are generated with a perturbation budget of 2/255, using 10 PGD steps and a step size of 0.5/255. **HQA-VLAttack (o, o)** means the model without the input transformation and contrastive learning. **HQA-VLAttack (o, w)** means the model without the input transformation, but with contrastive learning. **HQA-VLAttack (w, w)** means the model with input transformation and contrastive learning, i.e., the original model. The results are as follows, where higher values indicate better performance.
>
> | Method | ALBEF $\rightarrow$ $\text{CLIP}_{\text{ViT}}$ (TR R@1) | ALBEF $\rightarrow$ $\text{CLIP}_{\text{ViT}}$ (IR R@1) | ALBEF $\rightarrow$ $\text{CLIP}_{\text{CNN}}$ (TR R@1) | ALBEF $\rightarrow$ $\text{CLIP}_{\text{CNN}}$ (IR R@1) |
> | --- | --- | --- | --- | --- |
> | HQA-VLAttack (o,o) | 41.32 | 52.80| 46.49 | 57.63 |
> | HQA-VLAttack (o,w) | 49.08 | 59.50 | 55.94 | 63.57 |
> | HQA-VLAttack (w,w) | **52.15** | **62.05** | **59.64** | **65.59** |
>
> The results show that both the input transformation and the contrastive learning can affect the performance to some extent. We will add the above explanations in the final version.
>
> **3. About the contrastive learning.**
>
> As far as we know, HQA-VLAttack is the first work to incorporate the contrastive learning into a black-box multimodal adversarial attack.
>
> From the qualitative aspect, the contrastive loss can reduce the similarity between positive image-text pairs and increase the similarity between negative image-text pairs, i.e., it can guarantee that adversarial examples are pushed away from their true captions and pulled toward incorrect captions, thus improving the effectiveness of generated adversarial examples.
>
> From the quantitative aspect, we have conducted an ablation study for the contrastive learning and the input transformation, which is shown in the above Table. The results show that contrastive learning is very important for our method, which can improve the attack performance greatly.
>
> **4. The baselines incorporate the input transformation and the contrastive learning.**
>
> We would like to clarify that previous strong baselines have already used the input transformation in their original papers. So we just need to incorporate our proposed contrastive learning loss into their methods to verify the effectiveness. Specifically, we use ALBEF as the surrogate model on the Flickr30K dataset. Adversarial examples are generated with a perturbation budget of 2/255, using 10 PGD steps and a step size of 0.5/255. The results are shown as follows, where higher values indicate better performance.
>
>  | Method | ALBEF $\rightarrow$ $\text{CLIP}_{\text{ViT}}$ (TR R@1) | ALBEF $\rightarrow$ $\text{CLIP}_{\text{ViT}}$ (IR R@1) | ALBEF $\rightarrow$ $\text{CLIP}_{\text{CNN}}$ (TR R@1) | ALBEF $\rightarrow$ $\text{CLIP}_{\text{CNN}}$ (IR R@1) |
>  | --- | --- | --- | --- | --- |
>  | SGA | 33.38 | 44.16 | 34.93 | 46.57 |
>  | SGA+contrastive learning | 39.14 | 50.58 | 47.48 | 57.06 |
>  | DRA | 39.14 | 48.39 | 41.38 | 51.66 |
>  | DRA+contrastive learning | 43.19 | 57.38 | 49.17 | 58.46 |
>  | HQA-VLAttack | **52.15** | **62.05** | **59.64** | **65.59** |
>
> Based on the results, we can observe that contrastive learning can improve the baselines SGA and DRA greatly. In addition, HQA-VLAttack can consistently achieve the best performance, which reflects the effectiveness of the layer-wise importance score and the substitute word set in our method. We will add all the explanations in the final version.
>
> **5. About the computational cost.**
>
> (1) The runtime comparison of different methods is provided in Supplementary Material Section H (Table 2). From the results, it can be seen that HQA-VLAttack is a little slower than SGA, but much faster than DRA, which illustrates that the speed of HQA-VLAttack is acceptable.
>
> (2) We attempt to provide the computation complexity analysis as follows. Assume that $D$ is the feature dimension, $L$ is the text length, $C_{\text{mlm}}$ is the cost of one forward pass through a masked language model, $C_I$ and $C_T$ are the forward pass costs for the image encoder and text encoder, $G_I$ is the backward pass cost through the image encoder, $L_p$ is the number of image encoder layers, $D_p$ is the number of tokens per layer, $k$ is the number of substitute candidates per word, $T_p$ and $T_n$ are the number of positive and negative captions per batch, $S = |\text{Trans}(v'_i)|$ is the number of image scales used during contrastive learning, and $N$ is the number of attack iterations.
>
> **Text attack.**
>
> - For each word, we retrieve substitutes using either a counter-fitting synonym dictionary (constant-time) or an MLM ($C_{\text{mlm}}$). With the hit ratio $\rho$, the complexity of this step is $O(LC_{\text{mlm}}(1-\rho))$. We then evaluate $kL$ candidate texts by computing their similarity with a fixed image feature, yielding a total cost of $O(kL C_T)$. Therefore, the overall time complexity of the text attack is $O(LC_{mlm}(1−ρ)+kLC_T)$.
>
> **Image attack.**
> - Section 4.2.1 consists of determining layer importance and generating initial adversarial images. The first step requires one forward pass and $L_p$ vector comparisons. The second step needs to be repeated over $N$ iterations, including a forward pass, a backward pass, and layer-wise feature comparisons. Ignoring minor terms, the total complexity is $O(N(G_I + C_I + L_p D_p D))$.
>
> - Section 4.2.2 applies contrastive learning over multiple image scales. At each iteration, the adversarial image is compared with $T_p$ positive and $T_n$ negative texts across $S$ scales. This involves $S$ forward passes through the image encoder and $T_p + T_n$ forward passes through the text encoder, plus one backward pass. Ignoring minor terms, the total complexity is $O(N(SC_I + (T_p + T_n) C_T  + G_I))$.
> - Therefore, the total time complexity of the image attack is $O(N(G_I+C_I + L_p D_p D + SC_I + (T_p + T_n) C_T))$.
>
> Based on the above analysis, the total complexity for each image-text pair is $O(LC_{\text{mlm}}(1 - \rho) + kLC_T + N(G_I+C_I + L_p D_p D + SC_I + (T_p + T_n) C_T) )$.
>
> (3) We will add all the analysis in the final version.
>
> **6. About the minor suggestion.**
>
> The mentioned works offer useful ideas for improving the transferability of adversarial perturbations. We will add a more detailed discussion about them in the final version.
>
> **7. About the paper formatting concerns.**
>
> (1) For the appendix formatting, we follow the official Call for Papers of NeurIPS 2025. The original text is ''Technical appendices with additional results ....., or as a separate PDF in the ZIP file below before the supplementary material deadline''. So it meets the conference requirement.
>
> (2) For the code link, we use an anonymous GitHub account created with an anonymous email without any other information, thus meeting the double-blind review policy.

---

### Official Review · Reviewer_3F2f · 2025-07-01

**Clarity:** 3
**Significance:** 3
**Originality:** 3
**Rating:** 4
**Confidence:** 3

**Summary:**

This paper proposes HQA-VLAttack, a high-quality black-box adversarial attack method targeting vision-language pre-trained models. By jointly attacking both the textual and visual modalities, the method decreases the similarity of positive image-text pairs while increasing the similarity of negative pairs, thereby significantly improving the attack success rate. Experimental results demonstrate that HQA-VLAttack outperforms existing attack methods across multiple benchmark datasets, offering new perspectives and challenges for the security research of vision-language models.

**Questions:**

See above.

**Ethical Concerns:**

["NO or VERY MINOR ethics concerns only"]

**Final Justification:**

The author's response is well-written. I recommend that the authors include the additional discussion in the appendix. I maintain my original score.

**Limitations:**

Yes

**Quality:**

3

**Strengths And Weaknesses:**

Strengths:

1.The paper introduces the HQA-VLAttack framework. Unlike previous methods that focus solely on reducing the similarity of positive image-text pairs, this approach cleverly incorporates contrastive learning to increase the similarity of negative pairs. This bidirectional optimization strategy leads to more effective attacks.

2.The authors conduct experiments on three widely used multimodal benchmark datasets—Flickr30K, MSCOCO, and RefCOCO+—covering diverse task scenarios such as image-text retrieval, image captioning, and visual grounding. These experiments thoroughly validate the effectiveness of HQA-VLAttack across different tasks and datasets.

Weaknesses:

Overall, my experience reading this paper was quite positive. However, I do have a few concerns that need to be discussed:

1.MLLMs such as Qwen2.5-VL and LLaVA-Next are currently hot topics in the vision-language community. I suggest the authors include more discussion about these models, rather than just presenting two examples.

2.The number of iteration steps is set to 10. Could it be that some baseline methods have not fully converged due to the small number of steps, making the comparison potentially unfair?

---

> ### Author Rebuttal · Authors · 2025-07-31
>
> Thanks very much for your valuable and positive comments.
>
> **1. Discussion about Qwen2.5-VL and LLaVA-Next.**
>
> We have followed your comments to conduct additional experiments to evaluate the transferability of HQA‑VLAttack on Qwen2.5‑VL (`Qwen2.5-VL-32B-Instruct`)  and LLaVA-Next (`llava-v1.6-34b-hf`) under the image captioning task and visual question answering task. Specifically, we use ALBEF as the surrogate model to generate adversarial examples, with a perturbation budget of 16/255, 80 PGD steps, and a step size of 0.5/255.
>
> For the image captioning task, we randomly sample 128 image-text pairs from the Flickr30K dataset and generate adversarial images based on these clean pairs. Then we feed these adversarial images into Qwen2.5‑VL and LLaVA-Next to generate captions. The generated captions are evaluated with the widely-used metrics BLEU‑4, METEOR, ROUGE‑L, CIDEr, and SPICE, where lower values indicate better attack performance.
>
> For the visual question answering task, we randomly sample 128 examples from the VQAv2 dataset and apply perturbations to both the input images and their corresponding questions. The adversarial images and adversarial questions are then jointly fed into Qwen2.5‑VL and LLaVA-Next to generate answers. The generated answers are evaluated using the widely-used metric accuracy, where a lower accuracy indicates better attack performance. The detailed results are as follows.
>
> Note: The **Original** represents the performance on clean data.
>
> * **Image Captioning Results.**
>
> \> Attack Qwen2.5-VL.
>
>  | Method | BLEU‑4  | METEOR  | ROUGE-L  | CIDEr  | SPICE  |
>  | --- | --- | --- | --- | --- | --- |
>  | Original | 13.1 | 20.9 | 39.9 | 49.9 | 14.3 |
>  | SGA | 12.3 | 18.6 | 37.6 | 42.5 | 12.6 |
>  | DRA | 10.3 | 18.2 | 36.0 | 36.7 | 11.6 |
>  | **HQA-VLAttack** | **7.4** | **18.1** | **33.9** | **25.3** | **11.4** |
>
> \> Attack LLaVA-Next.
>
>  | Method | BLEU‑4  | METEOR  | ROUGE-L  | CIDEr  | SPICE  |
>  | --- | --- | --- | --- | --- | --- |
>  | Original | 14.1 | 15.7 | 33.2 | 42.5 | 10.6 |
>  | SGA | 5.8 | 13.8 | 30.1 | 30.8 | 9.9 |
>  | DRA | 5.8 | 13.3 | 28.8 | 28.7 | 9.5 |
>  | **HQA-VLAttack** | **4.8** | **11.9** | **27.8** | **23.7** | **7.8** |
>
> * **Visual Question Answering Results.**
>
>  | Method | Accuracy  (Qwen2.5-VL) | Accuracy  (LLaVA-Next) |
>  | --- | --- | --- |
>  | Original | 59.64 | 75.78 |
>  | SGA | 45.31 | 46.35 |
>  | DRA | 47.40 | 50.00 |
>  | **HQA-VLAttack** | **39.58** | **41.93** |
>
> These results show that HQA-VLAttack substantially degrades the performance on both Qwen2.5-VL and LLaVA-Next, which consistently outperforms SGA and DRA across all evaluation metrics. This further demonstrates the strong transferability and effectiveness of our method against modern MLLMs. We will add all these results in the final version.
>
> **2. About the iteration steps.**
>
> We have attempted to increase the iteration steps to 15, 20, and 25 for the strong baselines SGA and DRA on the Flickr30K dataset under the same transfer settings used in the original paper.
>
>
> Specifically, we use ALBEF as the surrogate model to attack $CLIP_{\text{ViT}}$ on both image‑to‑text retrieval (IR) and text‑to‑image retrieval (TR) tasks, and vice versa (i.e., using $CLIP_{\text{ViT}}$ as the surrogate model to attack ALBEF). We follow the original paper to use the attack success rate of R@1 as the evaluation metric for both IR and TR tasks, where a higher attack success rate indicates better attack quality. The detailed results are as follows.
>
> * **SGA Performance.**
>
>  | Steps | ALBEF → $\text{CLIP}_{\text{ViT}}$ (TR R@1) | ALBEF → $\text{CLIP}_{\text{ViT}}$ (IR R@1) | $\text{CLIP}_{\text{ViT}}$ → ALBEF (TR R@1) | $\text{CLIP}_{\text{ViT}}$ → ALBEF (IR R@1) |
>  | --- | --- | --- | --- | --- |
>  | 10 | 33.38 | 44.16 | 13.40 | 27.22 |
>  | 15 | 36.20 | 44.17 | 12.10 | 27.06 |
>  | 20 | 36.20 | 44.59 | 11.68 | 25.98 |
>  | 25 | 36.32 | 43.07 | 12.41 | 26.01 |
>
> * **DRA Performance.**
>
>  | Steps | ALBEF → $\text{CLIP}_{\text{ViT}}$ (TR R@1)  | ALBEF → $\text{CLIP}_{\text{ViT}}$ (IR R@1)  | $\text{CLIP}_{\text{ViT}}$ → ALBEF (TR R@1)  | $\text{CLIP}_{\text{ViT}}$ → ALBEF (IR R@1)  |
>  | --- | --- | --- | --- | --- |
>  | 10 | 39.14 | 48.39 | 12.51 | 30.00 |
>  | 15 | 39.14 | 48.74 | 13.45 | 29.51 |
>  | 20 | 38.40 | 48.32 | 14.70 | 29.98 |
>  | 25 | 38.90 | 48.74 | 14.08 | 29.44 |
>
> For the convenience of comparison, we also list the performance of HQA‑VLAttack with 10 iteration steps.
>
>  | Method | ALBEF → $\text{CLIP}_{\text{ViT}}$ (TR R@1) | ALBEF → $\text{CLIP}_{\text{ViT}}$ (IR R@1) | $\text{CLIP}_{\text{ViT}}$ → ALBEF (TR R@1) | $\text{CLIP}_{\text{ViT}}$ → ALBEF (IR R@1) |
>  | --- | --- | --- | --- | --- |
>  | HQA-VLAttack | **52.15** | **62.05** | **25.13** | **41.98** |
>
>
> Based on the above results, we can observe that increasing the number of steps does not lead to consistent performance improvement for SGA and DRA. And in some cases, the performance even slightly drops, which is probably due to the overfitting to the surrogate model. Moreover, HQA‑VLAttack with only 10 steps outperforms SGA and DRA even when their iteration steps are increased to 25. This indicates that our chosen step setting does not disadvantage the baseline methods and highlights the superior efficiency and effectiveness of HQA‑VLAttack. We will add all these results in the final version.

---

### Official Review · Reviewer_ZFZx · 2025-07-01

**Clarity:** 3
**Significance:** 3
**Originality:** 3
**Rating:** 4
**Confidence:** 4

**Summary:**

This paper proposes a novel transfer-based black-box adversarial attack on vision-language pre-trained models. The method combines three components: (1) counter-fitting word vectors for text adversarial generation, (2) layer importance-aware image perturbation, and (3) inclusion of negative pairs in image adversarial generation. It achieves state-of-the-art transferability across Image-Text Retrieval, Visual Grounding, and Image Captioning tasks.

**Questions:**

Question
- Does attack performance improve with a batch size larger than 16?
- Could you clarify the phrase “degrade the effectiveness of generating adversarial text” (line 131)? If the text is semantically inconsistent with the original word, wouldn’t that increase the chance of fooling the model by lowering image-text similarity?

**Ethical Concerns:**

["NO or VERY MINOR ethics concerns only"]

**Final Justification:**

After thorough discussion, the main concern was resolved.

**Limitations:**

Yes, in Appendix I and J.

**Quality:**

3

**Strengths And Weaknesses:**

Strength
- Incorporating negative pairs into adversarial image generation is a novel idea.
- The discussion on layer importance provides a valuable perspective.

Weakness
- *The evaluation protocol has a major flaw:* the proposed method assumes access to negative pairs from the test set, which are typically unavailable in real-world image-text retrieval systems. This makes the comparison unfair, as baseline methods only manipulate a single query and its paired sample, the proposed method presumes knowledge of additional samples in the database.
- The paper does not address how negative samples would be obtained in practice. Unlike existing methods that use only target image-caption pairs, the proposed method assumes access to a dataset, making its performance dependent on dataset distribution—an issue not discussed.
- The logic in Section 4.2.1 is unclear. The link between layer importance and adversarial transferability is not well established, and it’s uncertain whether important layers differ significantly across models. The proposed importance metric seems not well-justified.

---

> ### Author Rebuttal · Authors · 2025-07-31
>
> Thanks very much for your valuable and helpful comments.
>
> **1. About the evaluation protocol and the negative samples.**
>
> We would like to clarify the misunderstanding about the practicality of the evaluation protocol.
>
> (1) For negative pairs, we can construct them by randomly pairing each image with unrelated captions from the original dataset or any other external resources, which are easy to obtain in many real-world image-text retrieval systems. For example, in Google Image Search and TikTok platforms, the images and captions are publicly available and easy to access, which can be used to generate the negative pairs. So the evaluation protocol is practical in real-world scenarios.
>
> (2) To further verify our method does not rely on a specific dataset distribution and the evaluation protocol is practical, we conduct the following experiments. Specifically, we pair each adversarial image from the Flickr30K dataset with randomly sampled captions from the MSCOCO dataset. These captions are irrelevant to the source images and come from a totally different dataset. We use ALBEF as the surrogate model and attack against TCL, $CLIP_{\text{ViT}}$, and $CLIP_\text{CNN}$. We consider both image‑to‑text retrieval (IR) and text‑to‑image retrieval (TR) tasks. We follow the original paper to use the attack success rate of R@1 as the evaluation metric for both IR and TR tasks, where a higher attack success rate indicates better attack quality. For reference, the original results of SGA and DRA under the same evaluation setting are also included in the table below.
>
> |              | TCL (TR R@1) | TCL (IR R@1) | $\text{CLIP}_{\text{ViT}}$ (TR R@1) | $\text{CLIP}_{\text{ViT}}$ (IR R@1) | $\text{CLIP}_{\text{CNN}}$ (TR R@1) | $\text{CLIP}_{\text{CNN}}$ (IR R@1) |
> | ------------ | ------------ | ------------ | ----------------------------------- | ----------------------------------- | ----------------------------------- | ----------------------------------- |
> | SGA          | 45.42        | 55.25        | 33.38                               | 44.16                               | 34.93                               | 46.57                               |
> | DRA          | 49.74        | 58.83        | 39.14                               | 48.39                               | 41.38                               | 51.66                               |
> | HQA-VLAttack | **67.97**    | **75.12**    | **46.63**                           | **58.34**                           | **50.96**                           | **61.61**                           |
>
> From the results, we can get that HQA-VLAttack consistently outperforms other strong baselines even when negative captions are randomly sampled from a totally different dataset.
>
> (3) We will add all these explanations in the final version.
>
> **2. About the layer importance and adversarial transferability.**
>
> * Explanation of the design logic in Section 4.2.1.
>
> We would like to clarify the logic in Section 4.2.1, which consists of two parts: determining layer importance and generating initial image adversarial example. In the first part, we start by analyzing how different layers in the image encoder contribute to the final image representations. Specifically, we quantify each layer's influence by measuring the change in feature similarity when individual layers are skipped. This analysis allows us to define the importance for each layer based on the [CLS] token's cosine similarity to the final layer output. In the second part, we use these importance weights to guide the generation of initial adversarial images. We minimize the weighted sum of cosine similarities between the clean and perturbed image features across layers, where more important layers will receive higher weights, thus ensuring the transferability of the initial image adversarial example.
>
> * The link between layer importance and adversarial transferability.
>
> Intuitively, when the feature representations of the original example and adversarial example differ significantly, the adversarial transferability of generated perturbation is better. For the surrogate or target models, different layers usually contribute differently to feature representation learning. Therefore, just treating all layers equally seems unreasonable. To alleviate this issue, we propose to use importance scores to focus on some critical layers for feature representation learning, thus ensuring that the adversarial transferability of generated perturbation is satisfactory.
>
> * Whether important layers differ significantly across models.
>
> We have added the experiments to test whether important layers differ significantly across models. Specifically, we perform the analysis on two representative models TCL and $CLIP_{\text{ViT}}$. The importance scores of different layers are as follows.
>
> | Layer                          | 2    | 3    | 4    | 5    | 6    | 7    | 8    | 9    | 10   | 11   |
> | ------------------------------ | ---- | ---- | ---- | ---- | ---- | ---- | ---- | ---- | ---- | ---- |
> | $\textbf{CLIP}_{\textbf{ViT}}$ | 0.16 | 0.17 | 0.17 | 0.19 | 0.20 | 0.25 | 0.28 | 0.27 | 0.27 | 0.43 |
> | $\textbf{TCL}$                 | 0.82 | 0.79 | 0.80 | 0.83 | 0.81 | 0.84 | 0.82 | 0.91 | 0.92 | 0.94 |
>
> From the results, we can get that important layers do not differ significantly across models. Specifically, both models have a similar trend that higher layers seem more important, and the importance scores will sharply increase at a certain layer (7th layer for TCL and 9th layer for $CLIP_{\text{ViT}}$).
>
> * Analysis of importance metric.
>
> We have added the experiments to justify the effectiveness of the important score. Specifically, we use $CLIP_{\text{ViT}}$ as the surrogate model, remove importance weighting (i.e., setting all weights to 1), and attack against ALBEF. We consider both image-to-text retrieval (IR) and text-to-image retrieval (TR) tasks. We follow the original paper to use the attack success rate of R@1 as the evaluation metric for both IR and TR tasks, where a higher attack success rate indicates better attack quality. The details results are as follows.
>
> |                                      | TR R@1    | IR R@1    |
> | ------------------------------------ | --------- | --------- |
> | HQA-VLAttack (with importance score) | **25.13** | **41.98** |
> | HQA-VLAttack (no importance score)   | 24.09     | 40.87     |
>
> This confirms that the important score can improve the performance to some extent.
>
> We will add all the experiments and explanations in the final version.
>
>
> **4. About the batch size.**
>
> (1) In Figure 4(b) of Section 5.2.3 (in the original paper), we have provided the impact of different batch sizes {2, 4, 6, 8, 10, 12, 14, 16} on attack success rate.
>
> (2) We have added additional experiments to investigate the effect of batch size on attack performance. Specifically, we used ALBEF as the surrogate model to generate adversarial examples with a perturbation budget of 16/255, 80 PGD steps, and a step size of 0.5/255. We add the batch sizes to 18 and 22. The detailed results are as follows (the results with batch size is taken from the original paper).
>
> | Batch Size | TR R@1    | IR R@1    |
> | ---------- | --------- | --------- |
> | 16         | 73.02     | 77.60     |
> | 18         | 73.33     | 77.92     |
> | 22         | **73.96** | **78.44** |
>
> From the results, we can see that when the batch size increases beyond 16, the improvement of attack performance is slight. However, larger batch sizes will increase GPU memory consumption, so we set the batch size to 16 in our experiments.
>
>
> **5. About the phrase (line 131).**
>
> By "degrade the effectiveness of generating adversarial text," we mean that semantically inconsistent or unnatural texts will violate the definition of a high-quality adversarial example. Based on the widely accepted definition [1], a high-quality adversarial example should be human-imperceptible, i.e., the perturbation should be semantically consistent and unconspicuous. In other words, while extremely inconsistent text may increase the chance of fooling the model, it is easy to be observed. Hence, we aim to maintain the semantic consistency to ensure that adversarial texts are effective and human-imperceptible.
>
> [1] Intriguing properties of neural networks. ICLR 2014.

---

> > ### Comment · Reviewer_ZFZx · 2025-08-03
> > **Response by Reviewer**
> >
> > Thank you for your response.
> >
> > **1. About the evaluation protocol and the negative samples.**
> > - I do not think I have misunderstood anything here (please correct me if so). The evaluation protocol appears to have a critical flaw: if negative pairs are sampled from the *test set of the image-text retrieval task*, this directly influences the retrieval results and could be considered a form of data leakage (= cheating).
> > - Although the added results of Flickr30K-COCO are promising, they actually highlight that the attack success rate differs significantly when negative samples in database are not accessible.
> > - If I am correct, I believe that (1) how to construct/select negative pairs should be discussed and included in the part of the methodology, and (2) all evaluation tables should be revised to reflect a proper and realistic protocol.
> >
> > While some of my concerns were addressed, I still do not have sufficient confidence to recommend acceptance of this paper.

---

> > > ### Author Response · Authors · 2025-08-04
> > >
> > > Dear Reviewer ZFZx,
> > >
> > > **Thanks very much for your kind reply and insightful comments. We really happy to discuss with you.**
> > >
> > > (1) We would like to further clarify that there is no data leakage in this paper. The reasons are as follows. Data leakage means that a machine learning model accidentally learns from future data during training, thus leading to overly optimistic results and poor real-world performance. In contrast, for our black-box adversarial attack task, there is no train data, and the test set aims to generate the adversarial examples, not to make some prediction. Therefore, the data leakage is not tenable. In addition, our method is a transfer-based black-box attack method, which generate adversarial examples on a surrogate model and transfer them to deceive the victim model, i.e., we never query the victim model during generating the adversarial examples, thus further ensuring that data leakage is impossible.
> > >
> > > (2) We would like to further clarify the experimental settings of the transfer-based black-box attack. For example, take the Flickr30K dataset (1000 images and 5000 captions) as the test set, ALBEF as the surrogate model, and TCL as the victim model, we aim to leverage the ALBEF model to generate corresponding adversarial examples by using 1000 images and 5000 captions, and then use the generated adversarial examples to attack the TCL model. Finally, we evaluate the effectiveness of our method using the attack success rate on the TCL model. The above settings have been adpoted by the previous strong baselines like SGA and DRA, which can also access to the whole test set. That is to say, in the transfer-based black-box attack scenarios, no matter how to use the Flickr30K dataset and the surrogate model ALBEF, as long as the target model TCL is not accessed, the evaluation is reasonable. Based on the experiment settings, we think that data leakage is also not tenable.
> > >
> > > (3) For selecting the negative pairs, we think that both randomly pairing each image with unrelated captions from the original dataset or any other external resources are reasonable, the only requirement is not to access to the victim model, which is the key for the transfer-based black-box attack task.
> > >
> > > (4) In the above Flickr30K-COCO experiment, we sample only 4 negative captions from the MSCOCO dataset for each image due to the rebuttal time limitation. In contrast, in the original paper we use 75 negative captions per image. As a result, the reported performance in the above table is a little lower.
> > >
> > > To address your concerns, we have reconducted the experiments by randomly sampling 75 negative captions per image. Specifically, we pair each adversarial image from the Flickr30K dataset with randomly sampling 75 captions from the MSCOCO dataset. And we adopt ALBEF, TCL, $CLIP_{ViT}$, and $CLIP_{CNN}$ as surrogate models, and evaluate transferability to the remaining models. As in the original paper, we report attack success rates at R@1. For reference, we also include results of strong baselines under the same evaluation setting. The details results are as follows.
> > >
> > > \> Take ALBEF as the surrogate model.
> > >
> > > | |TCL (TR R@1)|TCL (IR R@1)|$\textbf{CLIP}_\textbf{ViT}$(TR R@1)|$\textbf{CLIP}_\textbf{ViT}$(IR R@1)|$\textbf{CLIP}_\textbf{CNN}$(TR R@1)|$\textbf{CLIP}_\textbf{CNN}$(IR R@1)|
> > > |-|-|-|-|-|-|-|
> > > |SGA|45.42|55.25|33.38|44.16|34.93|46.57|
> > > |DRA|49.74|58.83|39.14|48.39|41.38|51.66|
> > > |HQA-VLAttack|**71.44**|**77.02**|**51.90**|**62.34**|**59.49**|**65.87**|
> > >
> > > \> Take TCL as the surrogate model.
> > >
> > >  | |ALBEF (TR R@1)|ALBEF (IR R@1)|$\textbf{CLIP}_\textbf{ViT}$(TR R@1)|$\textbf{CLIP}_\textbf{ViT}$(IR R@1)|$\textbf{CLIP}_\textbf{CNN}$(TR R@1)|$\textbf{CLIP}_\textbf{CNN}$(IR R@1)|
> > >  |-|-|-|-|-|-|-|
> > >  |SGA|48.91|60.34|33.87|44.88|37.74|48.30|
> > >  |DRA|51.09|61.79|40.25|48.94|42.91|52.49 |
> > >  |HQA-VLAttack|**62.93**|**72.20**|**52.12**|**59.38**|**57.09**|**62.34**|
> > >
> > >  \> Take $CLIP_{ViT}$ as the surrogate model.
> > >
> > >  | |ALBEF (TR R@1)|ALBEF (IR R@1)|TCL(TR R@1) | TCL(IR R@1)|$\textbf{CLIP}_\textbf{CNN}$(TR R@1)|$\textbf{CLIP}_\textbf{CNN}$(IR R@1)|
> > >  |-|-|-|-|-|-|-|
> > >  |SGA|13.40|27.22|16.23|30.76|38.76|47.79|
> > >  |DRA|12.51|30.00|14.65|30.62|45.47|50.74|
> > >  |HQA-VLAttack|**25.23**|**42.04**|**24.24**|**43.45**|**74.46**|**77.23**|
> > >
> > >  \> Take $CLIP_{CNN}$ as the surrogate model.
> > >
> > >  | |ALBEF (TR R@1)|ALBEF (IR R@1)|TCL(TR R@1) | TCL(IR R@1)|$\textbf{CLIP}_\textbf{ViT}$(TR R@1)|$\textbf{CLIP}_\textbf{ViT}$(IR R@1)|
> > >  |-|-|-|-|-|-|-|
> > >  |SGA|11.42|24.80|14.91|28.82|31.24|42.12|
> > >  |DRA|12.20|26.59|14.33|29.29|35.21|45.94|
> > >  |HQA-VLAttack|**21.07**|**38.79**|**21.92**|**41.88**|**62.82**|**69.62**|
> > >
> > > We can observe that our method is relatively stable when the number of negative captions is set to 75, which further demonstrates our method does not rely on the dataset-specific distribution.
> > >
> > > **We will add all the explanations and experiments in the final version. Thanks very much again for your consideration.**
> > >
> > > Best regards,
> > >
> > > The Authors

---

> > > > ### Author Response · Authors · 2025-08-06
> > > >
> > > > Dear Reviewer ZFZx,
> > > >
> > > > We would like to thank you again for your insightful comments and hope that the new responses adequately address your concerns. We will follow your comments to add more explanations and discussion in the final version.
> > > >
> > > > Please let us know if you have any further questions or suggestions. Once again, we truly appreciate your time, effort, and thoughtful feedback.
> > > >
> > > > Best regards,
> > > >
> > > > Authors

---

> ### Comment · Reviewer_ZFZx · 2025-08-07
> **Response by Reviewer**
>
> Sorry for the late reply, and thank you for your detailed response.
>
> ## Regarding Evaluation Protocol
>
> Apologies for the confusion — I didn’t mean data leakage in the standard machine learning sense (i.e., training on future data).
> **What I meant is that HQA-VLAttack requires access to the database during the attack.**
>
> If I understand correctly, in an image-to-text retrieval setting where the attacker knows a positive pair (X, T):
> - Query: [X]
> - Database: [T, T2, T3, ..., Tn]
>
> SGA assumes access to (X, T) and uses their augmentations, which can be derived from the known pair.
> In contrast, **HQA-VLAttack seems to require knowledge not just of (X, T), but also the negative samples in the database (e.g., T4, T7, ...).** This allows the attacker to optimize the adversarial example X' such that:
> - cos-sim(X′, T) is minimized
> - cos-sim(X′, T4) is maximized
>
> — potentially causing X′ to match a specific negative sample T4 in the database, which should be inaccessible.
>
>
> ## Regarding Flickr30K experiment using 75 negative samples from COCO
> Sorry, I am comfused. I thought negatives were sampled within a batch size of 16 in the main paper due to computational constraints.

---

> > ### Author Response · Authors · 2025-08-08
> >
> > Dear Reviewer ZFZx,
> >
> > We would like to thank you again for your kind reply.
> >
> > **1. Regarding access to the database.**
> >
> > From the perspective of the attack objective, black-box adversarial attacks not only aim to simply attack a realistic system, but also to evaluate the robustness of the system. The attacker's goal is not to simulate extreme constraints, but to reveal whether the system can be fooled in practice. **Adversarial evaluation typically begins with a given dataset, from which adversarial examples are generated to assess the target model’s robustness. These examples are often reused for adversarial training to further enhance robustness. Therefore, allowing access to the database is a reasonable and widely accepted setting.**
> >
> > From the perspective of the practical applicatioin, many deployed systems expose their retrieval database publicly. For example, platforms like Google Image Search and TikTok provide access to large collections of image-caption pairs. **Attackers can easily use these public captions to construct negative pairs. Therefore, using the negative pairs is reasonable and practicable.**
> >
> > Moreover, HQA-VLAttack does not rely on the specific dataset. In our cross-dataset experiment, adversarial images are sampled from Flickr30K and paired with randomly sampled captions from MSCOCO. **Even without access to other negative captions from Flickr30K, the attack success rate of our method remains unaffected. This demonstrates the effectiveness of HQA-VLAttack even when no access to the target database is available.**
> >
> > We also note that SGA similarly requires access to the dataset during attack generation. **As stated in the original SGA paper (Section 4.3): *"we select the most matching caption pairs from the dataset of each image".*** This confirms that both SGA and our method assume dataset-level access when generating adversarial examples, making our protocol consistent with prior work.
> >
> > **2. Regarding the knowledge of negative pairs.**
> >
> > The use of negative pairs in our work is not only reasonable but also highly novel. In black-box attacks, the attacker does not have access to the target model’s internal parameters, gradients, or training dataset, and **there are no restrictions on the attack dataset under standard assumptions. Therefore, using negative sample pairs when generating adversarial examples does not violate any established assumptions of adversarial attack settings.**
> >
> > **One novelty of our method is explicitly increasing the similarity of negative pairs during adversarial example generation.** Prior methods (e.g., SGA, DRA) unintentionally reduce the similarity of both positive and negative pairs, limiting their attack ability. Our contrastive learning objective addresses this issue by better separating positive and negative pairs in the feature space, thus leading to better attack performance.
> >
> > **3. Regarding Flickr30K experiment using 75 negative samples from COCO.**
> >
> > Since each image is associated with 5 captions, a batch of 16 images contains a total of 80 captions. For each image, the 5 paired captions are excluded, leaving 75 remaining captions that can be used as negative pairs.
> >
> > Thank you again for your time and contribution to the review process. We really hope that the responses adequately address your concerns.
> >
> > Best regards,
> >
> > Authors

---

> > > ### Comment · Reviewer_ZFZx · 2025-08-08
> > > **Response by Reviewer**
> > >
> > > Thank you very much for your thorough response and the effort you put into addressing my concerns.
> > >
> > > Your explanations have clarified the key points.
> > > - Now I understand that some retrieval systems do expose their database, and that SGA similarly assumes data-level access to the victim system.
> > > - I suggest making this assumption clearer in the paper and including it in the revised version.
> > > - I also realize that accessing negative samples may not be difficult in practice, as one can simply query the system with random inputs.
> > >
> > > Nevertheless, it is still valuable to explore how the choice of negative samples affects the results in scenarios without database access. In particular, selecting negatives with lower similarity may further improve the attack success rate.
> > >
> > > Overall, my main concern has been resolved, and I will raise my score.
> > > Thank you again for your thoughtful response.

---

> > > > ### Author Response · Authors · 2025-08-08
> > > >
> > > > Dear Reviewer ZFZx,
> > > >
> > > > We are very pleased that our responses have addressed your concerns, and truly grateful for the increase in your assessment scores. We will include the explanations addressing your concerns in the final version of the manuscript. Your insightful feedback has been instrumental in helping us refine and strengthen our research. Thank you again for the time and effort you dedicated to reviewing our work.
> > > >
> > > > Best regards,
> > > >
> > > > Authors

---

### Note · Authors · 2025-08-12

Dear AC and SAC,

We sincerely want to express our gratitude for your dedication in reading through all the reviews and fostering meaningful discussions. The active participation of the reviewers greatly contributes to the enhancement of our paper’s quality and the enrichment of our contributions.

We really appreciate the reviewers’ recognition of the novelty and technical quality of our work. Based on the feedback of reviewers, we firmly believe that we have successfully addressed all the concerns raised by reviewers, including clarifications on methodology, additional experiments and theoretical explanations. We will incorporate all the discussions and clarifications into the final manuscript. And we truly hope that our work can make a meaningful contribution to the field.

Thanks very much again for your valuable time and generous consideration.

Best regards,

Authors

---

### Decision · Program_Chairs · 2025-09-17

**Decision:**

Accept (poster)

**Comment:**

This paper tries to improve the success rate of black-box attack on pretrained vision-language models. It proposes to use contrastive learning to increase the similarity of negative pairs, which is shown to be a very effective technique. Extensive reviewer–author interactions have helped resolve many key concerns and even persuaded some reviewers to raise their scores (or to reach positive final remarks). Considering the recommendations of all reviewers, the cleverness of the technique, the performance gain, and the importance of this topic, AC believes that this paper merits acceptance. The authors are required to include those important evidences/classifications from the rebuttal into the final paper. Also, since generating adversarial attacks might harm the security of pretrained models, the authors should discuss how to avoid malicious usage of this proposed technique.